# Coronary Malperfusion Secondary to Acute Type A Aortic Dissection: Surgical Management Based on a Modified Neri Classification

**DOI:** 10.3390/jcm11061693

**Published:** 2022-03-18

**Authors:** Guang Tong, Jinlin Wu, Zerui Chen, Donglin Zhuang, Shuang Zhao, Yaorong Liu, Yongchao Yang, Zhichao Liang, Ruixin Fan, Zhongchan Sun, Tucheng Sun

**Affiliations:** 1Department of Cardiac Surgery, Guangdong Cardiovascular Institute, Guangdong Provincial Key Laboratory of South China Structural Heart Disease, Guangdong Provincial People’s Hospital, Guangdong Academy of Medical Sciences, Guangzhou 510080, China; tongguang@gdph.org.cn (G.T.); wujinlin@gdph.org.cn (J.W.); chenzerui@gdph.org.cn (Z.C.); zhaoshuang@gdph.org.cn (S.Z.); liuyaorong@gdph.org.cn (Y.L.); yangyongchao@gdph.org.cn (Y.Y.); liangzhichao@gdph.org.cn (Z.L.); fanruixin@gdph.org.cn (R.F.); 2Department of Cardiovascular Surgery, Department of Structural Heart Disease, National Center for Cardiovascular Disease, China and Fuwai Hospital, Chinese Academy of Medical Sciences and Peking Union Medical College, Beijing 100037, China; rinrpg@student.pumc.edu.cn; 3Department of Cardiology, Guangdong Cardiovascular Institute, Guangdong Provincial Key Laboratory of South China Structural Heart Disease, Guangdong Provincial People’s Hospital, Guangdong Academy of Medical Sciences, Guangzhou 510080, China

**Keywords:** acute type A aortic dissection, coronary malperfusion, myocardial protection

## Abstract

Background: Coronary malperfusion (CM) secondary to acute type A aortic dissection (ATAAD) is considered rare but has a high mortality rate. This study examined the incidence, management, and outcomes of patients with CM secondary to ATAAD and proposes a modified Neri classification. Methods: Between 2015 and 2020, out of 1018 patients who underwent surgical repair for ATAAD, 137 presented with CM, including 68 (49.6%), 43 (31.3%), and 15 (10.9%) with Neri types A, B, and C, respectively, and 11 (8.0%) with coronary orifice intimal tear (COIT), which we consider a novel category. Results: The occurrence rate of CM was 13.4%. CM was associated with higher in-hospital mortality (18.2% vs. 7.8%, *p* < 0.001). For Neri type A (98.5%) and most type B lesions (72.1%), coronary repair was adequate. Coronary artery bypass grafting (CABG) was necessary for type B patients unsuited for repair (23.2%) and for all type C patients (100%). Repair of COIT was possible (45.5%). The in-hospital mortality rates differed significantly among the four lesion groups (*p* = 0.006). Conclusions: The occurrence of CM secondary to ATAAD may be more frequent than previously reported. Surgical management based on lesion classification achieved acceptable outcomes. Repair was adequate for Neri type A and most type B lesions. Other type B and type C lesions could be treated by CABG. Coronary orifice intimal tear is a unique set of lesions, for which orifice repair was also possible.

## 1. Introduction

Acute type A aortic dissection (ATAAD) is a deadly medical emergency with a high mortality rate [1]. Surgical repair remains the standard of care for patients with ATAAD [2]. Coronary malperfusion (CM) is a deadly complication of acute type A aortic dissection (ATAAD) [3], with a reported rate of 7% on postmortem examination [4]. Moreover, CM secondary to ATAAD is associated with a high postoperative mortality rate [5,6]. However, reports of occurrence, classification, and surgical management are scarce. Based on a study in 24 patients, CM following ATAAD has been categorized according to the Neri classification, with a management strategy tailored to each type [6]. Other case series have proposed different classification and surgical methodologies [5,7]. To date, however, the optimal treatment modality remains unclear, and no clear guidelines have been outlined. 

The present study reports the largest CM case series to date, consisting of 137 patients with CM. The aim of this study was to investigate the optimal management of patients with CM secondary to ATAAD, and to report our center’s classification, surgical approach, and outcomes for this patient cohort.

## 2. Materials and Methods

### 2.1. Patients

Of 1018 patients who underwent surgical repair for ATAAD from May 2015 to December 2020 at Guangdong Provincial People’s Hospital, 137 (13.4%) presented with preoperative CM. CM was diagnosed based on new ST-segment elevation detected by ECG, or significant elevation of serum cardiac enzymes in combination with enhanced CT imaging and intraoperative visualization of dissection involvement of the coronary artery [5,6]. Involvement of the coronary artery by dissection alone without clinical sign of cardiac ischemia was not considered as CM and was not included in the presented study.

### 2.2. Classification of CM

The patients were classified according to a modification of the Neri classification [6] (Figure 1), with (1) Neri A defined as ostial involvement by dissection; (2) Neri B as dissection extending into the actual body of the coronary artery (CA); (3) Neri C as circumferential detachment or complete avulsion of the CA. The fourth category, coronary orifice intimal tear (COIT), defined as the presence of an intimal tear in the area of the coronary ostium without avulsion of the CA, is considered a novel, independent category because of its unique pathology and management strategy.

### 2.3. Surgical Technique

Our standard surgical technique included median sternotomy, proximal aortic repair under cardiopulmonary bypass (CPB), and open distal aortic repair under hypothermic circulatory arrest (HCA) with selective cerebral perfusion (SCP). Proximal aortic repair techniques included supracoronary aortic replacement, sinus of Valsalva repair with bovine pericardium neomedia, aortic valve resuspension, aortic root replacement with a valved conduit, aortic valve replacement (AVR) plus supracoronary aortic replacement), or valve-sparring aortic root replacement. Aortic root replacement was performed in patients with a dilated aortic root (>45 mm), extensive root destruction, or connective tissue disease.

A 4-branched graft (Hemashield Platinum Branch Graft, Maquet, Rastatt, Germany; or Gelweave Plexus Graft, Terumo, Tokyo, Japan) was used for arch reconstruction. Separate graft or en-bloc technique were used for arch vessels reconstruction. Frozen elephant trunk (FET; MicroPort Medical Co. Ltd., Shanghai, China) was routinely performed to avoid future reintervention. 

### 2.4. Myocardial Protection and CA Repair

Both antegrade and retrograde cardioplegia were used. The CA were carefully examined before the application of antegrade cardioplegia. If the dissected CA was deemed unsuitable for the safe application of antegrade cardioplegia, retrograde cardioplegia was used, and antegrade cardioplegia was applied exclusively to the undissected CA. If the CA was too extensively damaged for repair (e.g., unrepairable orifice intimal tear, avulsion of the CA, collapse of the CA true lumen by false lumen thrombosis) and CABG was deemed necessary, a saphenous vein graft (SVG) was promptly harvested and anastomosed to the CA, and the vein graft was used for further antegrade cardioplegia administration. 

The strategy of CA repair was largely dictated by the lesion type. A reparative strategy, consisting of ostia or supracoronary repair, was successfully applied to the majority of patients with type A lesions. For ostia repair, the coronary ostium was repaired with continuous 6-0 over-and-over sutures conjoining the dissected arterial layers [6]. Supracoronary repair was mostly performed for dissection involvement limited to the superior portion of the orifice. The proximal aortic stump was reinforced at the sinotubular (ST) junction with continuous 6-0 horizontal mattress plus over-and-over sutures and fortified with bovine pericardium strips on both the inner and the outer walls of the aorta, if deemed necessary. For type B lesions, the extent of the dissection was carefully determined. The surface of the heart was examined for peri-coronary hematoma secondary to coronary dissection. Antegrade cardioplegia was applied with a slow increase in flow pressure to detect potential CA compression. Orifice or supracoronary repair was the first choice for CA repair of type B lesions. After release of the cross clamp, the heart was carefully inspected. If signs of repair failure, such as poor myocardial contraction or expansion of the peri-coronary myocardial hematoma, were detected, then the patient underwent immediate conversion to CABG. Primary CABG was performed if the orifice tissue was too fragile or if CA compression was considered likely after repair. In these patients, the orifice of the CA was oversewn with running 6-0 prolene sutures. All type C lesions underwent CABG. COIT was considered a unique set of lesions. Each intimal tear was carefully examined, as were root damage and anatomy. If the intimal tear was limited and the intima sufficiently resilient to hold stitches, then intimal repair, juxtaposing the layers together, was performed with continuous 6-0 over-and-over sutures. Otherwise, CABG was performed, and the orifice of the CA was oversewn. 

### 2.5. Clinical Endpoints

In-hospital mortality, ECMO use, and new stroke were primary outcomes of interest. The remaining adverse events including revisiting for bleeding, mediastinitis, paraplegia, continuous renal replacement, tracheostomy were considered secondary outcomes.

### 2.6. Statistical Analysis

Continuous data were evaluated for normality using the Kolmogorov–Smirnov test. Normally distributed data were expressed as mean ± standard deviation (SD) and compared by one-way ANOVA, whereas abnormally distributed data were expressed as median with interquartile range (IQR) and compared by the Kruskal–Wallis test. Categorical variables were reported as frequencies with percentages and compared by the chi-square test or Fisher’s exact test. 

A two-tailed *p* value < 0.05 was considered statistically significant. Statistical analyses were performed using IBM SPSS Statistics (IBM, Armonk, NY, USA).

## 3. Results

### 3.1. Pre- and Intraoperative Characteristics of the Entire Cohort

The pre-operative characteristics of the whole cohort are summarized in Table 1. Cardiogenic shock (29.9% vs. 9.8%, *p* < 0.001) and coronary heart disease (CAD) (11.7% vs. 6.7%, *p* = 0.038) were more frequently present in patients with than in those without CM. 

The intra-operative characteristics are summarized in Table 2. Root replacement (56.9% vs. 31.0%, *p* < 0.001), V-SARR (8.0% vs. 5.0%, *p* = 0.013), and CABG (26.3% vs. 3.6%, *p* < 0.001) were performed more frequently in patients with in those than without CM. Both CPB time (273.6 ± 78.8 vs. 241.8 ± 63.5, *p* < 0.001) and aortic cross-clamp time (151.1 ± 48.5 vs. 128.2 ± 42.3, *p* < 0.001) were longer in patients with than in those without CM. 

### 3.2. Outcomes of the Entire Cohort

In the entire population, 94 patients (9.2%) died during hospital stay. The mortality rate was higher in the CM group (18.2% vs. 7.8%, *p* < 0.001). The incidence of ECMO use (10.9% vs. 2.3%, *p* < 0.001), paraplegia (5.8% vs. 1.7%, *p* = 0.002), CRRT use (29.9% vs. 20.5%, *p* = 0.013) were also higher in the CM group (Table 3).

### 3.3. Pre- and Intraoperative Characteristics of the CM Patients

Of the entire cohort of 137 patients with CM, 41 (29.9%) presented with cardiogenic shock, 23 (16.8%) with cardiac tamponade, and 36 (26.3%) with moderate to severe aortic insufficiency. In addition, 39 (28.5%) patients presented with malperfusion of other systems.

The side of the affected CA did not differ significantly in the patient cohort, although the right CA was predominantly affected in all four lesion types. Analysis of the patients with type A lesions showed that the left CA alone was affected in 5 (7.4%) patients, the right CA alone in 45 (66.2%) patients, and both the left and the right CA in 18 (26.5%) patients. Among patients with type B lesions, the left CA alone was affected in 4 (9.3%) patients, the right CA alone in 32 (74.4%) patients, and both the left and the right CA in 7 (16.3%) patients. Among patients with type C lesions, the right CA alone was affected in 12 (80.0%) patients, whereas both the left and the right CA were affected in 3 (20.0%) patients. In the COIT group, the left CA alone was affected in 1 (9.1%) patient, and the right CA alone was affected in 10 (90.9%) patients. Overall, the right CA was more frequently affected than the left CA, with bilateral CA malperfusion present in 28 patients (20.4%) (Table 4). 

The intra-operative characteristics of the CM patients are summarized in Table 5. Coronary repair differed significantly between the four lesion types. Of the 68 type A lesions, 67 (98.5%) were successfully repaired, with orifice repair performed in 46 (67.6%) and ST junction repair in 21 (30.9%) lesions. Of the 43 type B lesions, 31 (72.1%) were successfully repaired, including in 21 (48.8%) which underwent successful orifice repair, and 10 (23.3%) which underwent successful ST junction repair. Thirteen (30.2%) patients in this group underwent primary CABG. All 15 type C lesions underwent primary CABG. Of the 11 patients with COIT, 5 (45.5%) underwent successful orifice repair, and 6 (54.5%) underwent primary CABG. For proximal repair, the aortic root had to be replaced more frequently in patients with type C (86.7%) and COIT (81.8%) lesions than in patients with type A (45.6%) and type B (58.1%) lesions (*p* = 0.008). There were no differences in distal aortic repair between the four lesion types (Table 5). 

### 3.4. In-Hospital Outcomes of the CM Patients

The outcome characteristics of the CM patients are reported in Table 6 and Figure 2. Fifteen (10.9%) patients required postoperative ECMO support. The in-hospital mortality rates differed significantly between patients in the type A (7.4%), type B (25.6%), type C (40.0%), and COIT (27.3%) groups (*p* = 0.006). The incidence of stroke was also significantly higher in the type C (13.3%) and COIT (27.3%) groups than in the type A (5.9%) and type B (2.3%) groups (*p* = 0.028). Patients in the type C group also required ECMO support (26.7%) more often than patients in the type A (4.4%), type B (16.7%), and COIT (9.1%) groups (*p* = 0.045). The incidences of composite adverse events did not differ significantly between the type A, type B, type C, and COIT groups (36.8% vs. 44.2% vs. 66.7% vs. 54.5%, *p* = 0.167).

## 4. Discussion

The essential findings of the presented study are as follows: (1) CM occurred in 13.5% of ATAAD patients who survived to surgery, more frequently than described by previous reports; (2) an individualized surgical strategy based on lesion classification is vital for optimal outcomes. Repair was adequate for coronary repair of Neri type A and most type B lesions. CABG is a simple and straightforward revascularization technique for type C lesions; (3) COIT is a unique set of lesions, and orifice repair was possible for COIT. 

The occurrence of CM is considered relatively rare, as a postmortem study reported an occurrence of 7% [4]. Surgical case series are sparse, and the reported incidence of CM among ATAAD patients was 6.1% [7], 9% [5], and 11.3% [6] in different studies. In our series, CM occurred in 13.5% of ATAAD patients, a higher rate than those reported in these previous studies. One possible explanation may be the relative younger age of our patient cohort (50.3 ± 11 years), as coronary involvement following ATAAD has been associated with younger age [8]. In patients who survive to surgery, the right CA is much more often affected than the left CA, with the latter affected alone only in 7.3% of patients. The higher incidence of right CA dissection is attributed to the fact that a false lumen develops most often in the right anterior aspect of the ascending aorta [9]. Type C lesions occurred exclusively in the right CA. This clearly resulted from a severe survival bias, as most patients with type C lesions and complete avulsion of the left main stem would likely die before reaching a hospital in time for adequate diagnosis [5]. The actual occurrence of CM would be even higher than detected among surgical patients.

CM secondary to ATAAD can be caused by either static or dynamic obstruction of coronary blood flow. Expeditious restoration of coronary blood flow to all parts of the myocardium is vital for patient survival [2]. The ideal strategy to restore blood flow in the malperfused myocardium remains controversial. Neri et al. [6] suggest direct CA repair over CABG. Kawahito et al. [7], however, recommend using CABG in all patients with CA dissection. Mobilization and repair of frail coronary ostia buttons can potentially fail. On the other hand, CABG always carries the risk of graft failure.

In our opinion, rather than a universal therapeutic rule, a strategy based on the pathologic cause of CA malperfusion should be applied. Optimal surgical repair demands lesion classification. We found that almost all type A lesions (98.5%) could be successfully repaired, in agreement with previous reports [5,6]. Orifice repair is the surgical treatment of choice for patients with these lesions. In addition to orifice repair, these patients can also undergo supracoronary repair, as ATAAD sometimes involves only the upper part of the coronary orifice. CABG would only be necessary in case of evidence of CAD. 

The repair of type B lesions can be tricky. Longitudinal incision and patch repair of the affected CA have been advocated for type B lesions by Neri et al. [6]. Of the 10 patients with type B lesions in that study, 7 underwent RC repair, and 3 underwent LC repair, with all 10 surviving. Patch repair, however, is an invasive and time-consuming technique, requiring profound dissection of edematous, fragile tissue along an unclear length of the CA to reach the non-dissected portion of the CA tissue. In our series, 72.1% of patients with type B lesions (excluding ostia intimal tear) underwent successful coronary repair. Similarly, another study reported that 20 of 32 patients with type B lesions underwent successful ostia repair by retacking the layers back together in a sandwich-like fashion with pledgetted sutures [5]. Thus, simple ostia or supracoronary repair was our favored technique for type B lesions. In some patients, however, CABG with dissected ostia closure was necessary, as the ostia tissue was too fragile for resilient repair, or the true lumen of the CA was compressed by a false lumen thrombus. When repairing type B lesion, the surgeon should be prepared for repair failure. One out of 32 (3%) repairs failed in our study. Kreibich et al. [5] reported 3 failures out of 23 repairs for type B lesions (13%). Failure of orifice repair required salvage CABG.

Because the simple repair of type C lesions is not possible, transection of the CA in a nondiseased zone and reconstruction of the vessel with SVG using end-to-end anastomosis were proposed [6]. Conventional CABG in the non-dissected portion of the CA is a simpler alternative and has been successful [5,8]. In our series, all type C lesions occurred in the right CA, making CABG straightforward and much less surgically challenging. CABG to the right CA resulted in good graft flow in all 15 type C patients. Although these findings suggest that CABG is a simple and effective way for right CA type C lesions, the optimal strategy for left CA type C lesions remains elusive. 

No COIT was described in Neri report [6]. Chen et al. reported six cases of disruption of the inner layer limited to the area of the coronary ostium and considered it as a unique lesion set [8]. Five cases of entry tear at the coronary ostia were described by Kreibich et al. and considered as Neri B lesion [5]. Pathologically, they can be considered as partial type C lesions without coronary intussusception. In both reports, universal sacrifice of the ostia and CABG was thought to be necessary for all lesions of this type [5,8]. In our opinion, repair was possible. The intimal tear should be carefully examined, along with the root damage and anatomy. If the intimal tear is limited and the intima is resilient enough to hold stitches, then intimal repair with continuous 6-0 over-and-over sutures rejuxtaposing the layers together is possible. Repair was performed with success in 5 of the 11 patients in the COIT group. 

In our cohort, the mortality rate of the CM patients was 18.2%, significantly higher than that of patients without CM but comparable to that of previous reports [5,6]. However, the mortality rates in the four groups were significantly different. The mortality rate of patients with type A lesions was 7.4%, similar to that of non-CM patients, whereas the mortality rates of patients with type B (25.6%), type C (40.0%), and COIT (27.3%) lesions were significantly higher (*p* = 0.006). A more extensive CM involvement was likely to be associated with more severe and extensive myocardial ischemic injury. Severe root destruction associated with extensive CM involvement increased the requirement for more complicated root procedures. The rate of root replacement was significantly higher for patients in the type C (86.7%) and COIT (81.8%) groups than for the type A (45.6%) and type B (58.1%) groups (*p* = 0.008), with patients in the type C and COIT groups requiring longer aortic cross clamp time and CPB time, which are associated with poorer outcomes. 

The present study has a few inherent limitations. The sample size was small, and the study was retrospective in design. All patients were treated at a single center. Multi-centered, large sample studies are required in the future to optimize the surgical management for these patients. 

## 5. Conclusions

The occurrence of CM secondary to ATAAD may be more frequent than previously reported, especially among younger patients. A surgical strategy based on lesion classification is vital for optimal outcomes. Orifice or supracoronary repair was adequate for coronary repair of Neri type A and most type B lesions. Patients with fragile orifice tissue or apparent peri-coronary hematoma should undergo CABG. CABG is a simple and straightforward revascularization technique for type C lesions. COIT is a unique set of lesions, and its management should be individualized. Orifice repair was possible for COIT.

## Figures and Tables

**Figure 1 jcm-11-01693-f001:**
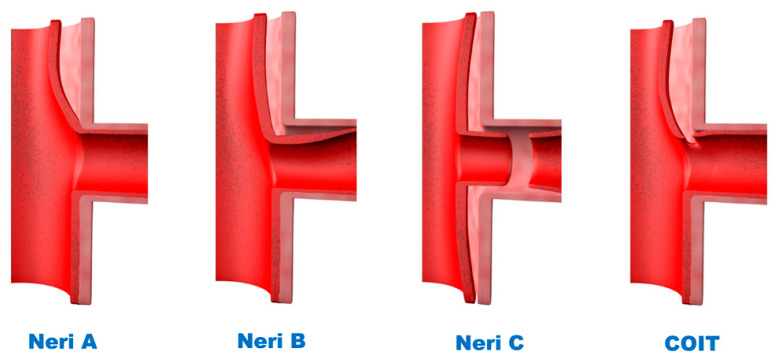
A modified Neri Classification of CM due to ATAAD. CM, coronary malperfusion; ATAAD, acute type A aortic dissection; COIT, coronary ostium intimal tear.

**Figure 2 jcm-11-01693-f002:**
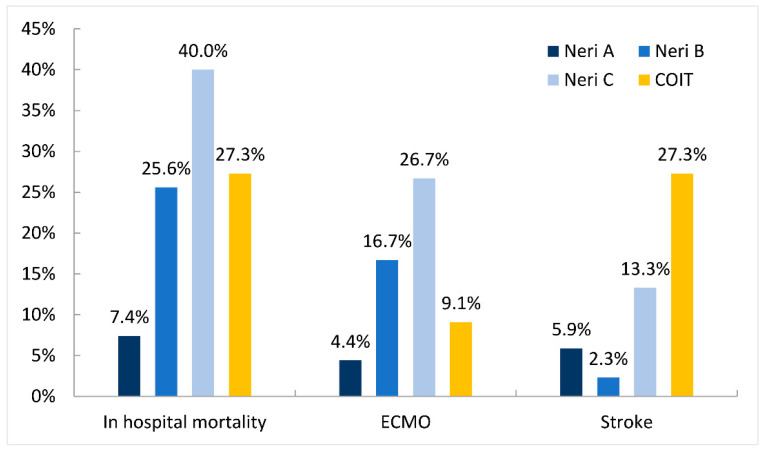
In-hospital mortality, ECMO use, and postoperative stroke for each lesion type. ECMO, extracorporeal membranous oxygenation; COIT, coronary ostium intimal tear.

**Table 1 jcm-11-01693-t001:** Preoperative characteristics of the entire cohort.

Variables	Total (*n* = 1018)	CM (*n* = 137)	No CM (*n* = 881)	*p* Value
Demographic				
Age (mean ± SD)	50.5 ± 10.8	50.3 ± 11	50.5 ± 10.7	0.799
Male (%)	855 (84.0)	116 (84.7)	739 (83.9)	0.815
Cardiogenic shock(%)	127 (12.5)	41 (29.9)	86 (9.8)	<0.001
Tamponade/blood effusion	122 (12.0)	23 (16.8)	99 (11.2)	0.067
Hypertension (%)	617 (60.0)	93(67.9)	524 (59.5)	0.061
Connective tissue disorder	46 (4.5)	5(3.6)	41 (4.7)	0.825
Diabetes mellitus (%)	23 (2.3)	3(2.3)	20 (2.3)	1.000
Smoking (%)	182 (17.9)	30 (21.9)	152 (17.3)	0.187
History of stroke (%)	60 (5.9)	8 (5.8)	52 (5.9)	0.977
Coronary heart disease (%)	75 (7.4)	16 (11.7)	59 (6.7)	0.038
Chronic renal dysfunction (%)	20 (2.0)	3 (2.2)	17 (1.9)	0.743
History of heart/aortic surgery (%)	50 (4.9)	8 (5.8)	42 (4.8)	0.589
Atrial fibrillation (%)	9 (0.9)	1 (0.7)	8 (0.9)	1.000
COPD (%)	18 (1.8)	2 (1.5)	16 (1.8)	1.000
Malperfuison				
Cerebral (%)	46 (4.5)	4 (2.9)	42 (4.8)	0.505
Renal (%)	201 (19.7)	28 (20.4)	173 (19.6)	0.827
Gastrointestinal (%)	60 (5.9)	7 (5.1)	53 (6.0)	0.675
Iliofemoral (%)	39 (3.8)	2 (1.5)	37 (4.2)	0.152
Spinal (%)	14 (1.4)	2 (1.5)	12 (1.4)	1.000

COPD, chronic obstructive pulmonary disease.

**Table 2 jcm-11-01693-t002:** Operative characteristics of the entire cohort.

Variables	Total (*n* = 1018)	CM (*n* = 137)	No CM (*n* = 881)	*p* Value
Proximal repair				
Supracoronary aortic replacemnt (%)	311 (30.6)	16 (11.7)	295 (33.5)	<0.001
Commissure suspension (%)	249 (24.5)	22 (16.2)	227 (25.8)	0.015
AVR + supracoronary aortic replacement (%)	16 (1.6)	3 (2.2)	13 (1.5)	0.465
Root replacement (%)	351 (34.5)	78 (56.9)	273 (31.0)	<0.001
sinus replacement with neomedia (%)	37(3.6)	8 (5.8)	29 (3.3)	0.138
V-SARR (%)	55 (5.4)	11 (8.0)	44 (5.0)	0.013
CABG (%)	62 (7.3)	36 (26.3)	26 (3.6)	<0.001
Arch vessels reconstruction, *n* (%)				
branched graft (%)	694 (68.2)	119 (86.9)	575 (65.3)	<0.001
En-bloc (%)	323 (31.7)	18 (13.1)	305 (34.6)	<0.001
FET (%)	999 (98.1)	137 (100)	862 (97.8)	0.083
Time/Temperature				
CPB time (mean ± SD)	246.1 ± 66.6	273.6 ± 78.8	241.863.5	<0.001
Aortic cross-clamp time (mean ± SD)	131.3 ± 43.9	151.1 ± 48.5	128.242.3	<0.001
HCA time (mean ± SD)	22.1 ± 8.4	21.2 ± 7.3	22.2 ± 8.6	0.198
Lowest HCA temperature (mean ± SD)	22.5 ± 3.4	22.3 ± 2.7	22.5 ± 3.5	0.443

V-SARR, valve-sparing aortic root replacement; CABG, coronary artery bypass grafting; FET, frozen elephant trunk; CPB, cardiopulmonary bypass; HCA, hypothermic circulatory arrest.

**Table 3 jcm-11-01693-t003:** Outcomes of the entire cohort.

Variables	Total (*n* = 1018)	CM (*n* = 137)	No CM (*n* = 881)	*p* Value
In hospital mortality (%)	94 (9.2)	25 (18.2)	69 (7.8)	<0.001
ECMO (%)	35 (3.4)	15 (10.9)	20 (2.3)	<0.001
New stroke (%)	67 (6.6)	10 (7.3)	57 (6.5)	0.716
Revisiting for bleeding (%)	90 (8.8)	17 (12.4)	73 (8.3)	0.114
Mediastinitis (%)	11 (1.1)	2 (1.5)	9 (1.0)	0.650
Paraplagia (%)	23 (2.3)	8 (5.8)	15 (1.7)	0.002
CRRT (%)	222 (21.8)	41 (29.9)	181 (20.5)	0.013
Tracheostomy (%)	31 (3.0)	6 (4.4)	25 (2.8)	0.329
Times				
Ventilation time, d, (median [IQR])	4.0 [2.0, 7.0]	5.0 [3.0, 7.5]	4.0 [2.0, 7.0]	0.200
ICU stay, d, (median [IQR])	8.0 [5.0, 13.0]	9.0 [6.0, 15.0]	8.0 [5.0, 12.0]	0.119
Hospital stay, d, (median [IQR])	22.0 [17.0, 30.0]	21.0 [14.0, 30.0]	22.0 [17.0, 30.0]	0.147

ECMO, extracorporeal membrane oxygenation; CRRT, continuous renal replacement therapy; ICU, intensive care unit.

**Table 4 jcm-11-01693-t004:** Preoperative characteristics of patients with CM.

Variables	Overall (*n* = 137)	Neri A (*n* = 68)	Neri B (*n* = 43)	Neri C (*n* = 15)	COIT ^a^ (*n* = 11)	*p* Value
Demographic						
Age, year (median [IQR])	51.0 [45.0, 57.0]	52.5 [46.0, 57.0]	50.0 [41.0, 56.0]	51.0 [48.0, 57.0]	56.0 [39.0, 63.0]	0.747
Male (%)	116 (84.7)	56 (82.4)	38 (88.4)	13 (81.8)	9 (81.8)	0.837
Hypertension (%)	93 (67.9)	47 (69.1)	34 (79.1)	8 (53.3)	4 (36.4)	0.029
Connective tissue disorder (%)	5 (3.6)	2 (2.9)	1 (2.3)	0 (0.0)	2 (18.2)	0.058
Smoking (%)	107 (78.1)	56 (82.4)	31 (72.1)	10 (66.7)	10 (90.9)	0.281
CAD (%)	16 (11.7)	7 (10.3)	6 (14.0)	0 (0.0)	2 (27.3)	0.178
CA malperfusion						
Isolated left (%)	10 (7.3)	5 (7.4)	4 (9.3)	0 (0.0)	1 (9.1)	0.685
Isolated right (%)	99 (72.3)	45 (66.2)	32 (74.4)	12 (80.0)	10 (90.9)	0.294
Bilateral (%)	28 (20.4)	18 (26.5) ^b^	7 (16.3) ^c^	3 (20.0) ^d^	0 (0.0)	0.187
Other system malperfusion						
Cerebral (%)	4 (2.9)	2 (2.9)	2 (4.7)	0 (0.0)	0 (0.0)	0.744
Renal (%)	28 (20.4)	17 (25.0)	5 (11.6)	5 (40.0)	0 (0.0)	0.026
Gastrointestinal (%)	7 (5.1)	4 (5.9)	3 (7.0)	0 (0.0)	0 (0.0)	0.616
Iliofemoral (%)	2 (1.5)	1 (1.5)	1 (2.3)	0 (0.0)	0 (0.0)	0.894
Spinal (%)	2 (1.5)	2 (2.9)	0 (0.0)	0 (0.0)	0 (0.0)	0.560
Shock/Hypotension (%)	41 (31.5)	20 (29.4)	10 (23.3)	7 (46.7)	4 (36.4)	0.370
Tamponade (%)	23 (16.8)	13 (19.1)	5 (11.6)	3 (20.0)	2 (18.2)	0.751
Moderate to severe AI (%)	36 (26.3)	18 (26.5)	10 (23.3)	4 (26.7)	2 (36.4)	0.854

CAD, coronary artery disease; CA, coronary artery; AI, aortic insufficiency. ^a^ Right CA orifice intimal tear in 10 patients, left CA orifice intimal tear in 1 patient. ^b^ Bilateral type A lesions. ^c^ Both type B lesions in six patients, right CA type B+ left CA type A in one patient, patients not included in the type A population. ^d^ Right CA type C+ left CA type A in one patient, right CA type C+ left CA type B in two patients, patient not included in the type A or type B population.

**Table 5 jcm-11-01693-t005:** Operative characteristics in patients with CM.

Variables	Overall (*n* = 137)	Neri A (*n* = 68)	Neri B (*n* = 43)	Neri C (*n* = 15)	COIT (*n* = 11)	*p* Value
Coronary repair, *n* (%)						
Successful repair	105 (76.6)	67 (98.5)	31 (72.1)	2 (13.3)	5 (45.5)	<0.0001
Successful coronary orifice repair	74 (54.0)	46 (67.6)	21 (48.8) ^b^	2 (13.3) ^a^	5 (45.5)	<0.0001
Successful ST junction repair	31 (22.6)	21 (30.9)	10 (23.3)	0 (0.0)	0 (0.0)	0.016
Failed repair/CABG conversion	1 (0.7)	0 (0.0)	1 (2.3) ^c^	0 (0.0)	0 (0.0)	0.532
Primary CABG	35 (25.5)	1 (1.5)	13 (30.2)	15 (100.0)	6 (54.5)	<0.0001
Proximal repair, *n* (%)						
Supracoronary aortic replacement	16 (11.7)	8 (11.8)	7 (16.3)	1 (6.7)	0 (0.0)	0.440
Sinus replacement with neomedia	8 (5.8)	4 (5.9)	4 (9.3)	0 (0.0)	0 (0.0)	0.466
Aortic valve resuspension	21 (15.3)	19 (27.9)	2 (4.7)	0 (0.0)	0 (0.0)	<0.0001
Aortic root replacement	78 (56.9)	31 (45.6)	25 (58.1)	13 (86.7)	9(81.8)	0.008
AVR + supracoronary aortic replacement	3 (2.2)	1 (1.5)	1 (2.3)	1 (6.7)	0 (0.0)	0.611
V-SARR	11 (8.0)	5 (7.4)	4 (9.3)	0 (0.0)	2 (18.2)	0.394
Arch vessels reconstruction, *n* (%)						
branched graft (%)	18 (13.1)	6 (11.8)	7 (16.3)	2 (13.3)	1 (9.1)	0.887
En-bloc (%)	119 (86.9)	60 (88.2)	36 (83.7)	13 (86.7)	10 (90.9)	0.887
FET	137 (100)	66 (100)	44 (100)	20 (100)	7 (100)	-
Time/Temperature						
CPB time, min (mean ± SD)	273.6 ± 78.8	243.2 ± 46.7	298.0 ± 95.2	297.6 ± 86.6	333.2 ± 88.0	<0.0001
ACC time, min (median [IQR])	151 [117, 175]	138 [114.25, 164.75]	158 [126, 196]	157 [116, 184]	175 [138, 199]	0.024
HCA time, min (median [IQR])	21 [17, 25]	20 [17, 26.75]	21 [18, 25]	20 [15, 23]	22 [19, 25]	0.837
Lowest HCA temperature, °C (median [IQR])	22.5 [20.55, 24]	22.6 [20.625, 23.8]	22.6 [20.9, 24.3]	21.8 [18.7, 23.6]	22.5 [20.5, 24.3]	0.697

CABG, coronary artery bypass grafting; V-SARR, valve-sparing aortic root replacement; CPB, cardiopulmonary bypass; ACC, aortic cross clamp; HCA, hypothermic circulatory arrest. ^a^ One patient with an RC type C and an LC type A lesion, one patient with an RC type C and an LC type B lesion; both LC lesions underwent orifice repair, both RC lesions were repaired by primary CABG. ^b^ Two patients with RC type B and LC type A lesions; both RC type B lesions were repaired with CABG, and both LC type A lesions underwent orifice repair. ^c^ One patient with an RC type B and an LC type B lesion; the RC type B lesion was successfully repaired with CABG, whereas the orifice repair for the LC type B lesion failed and was converted to CABG.

**Table 6 jcm-11-01693-t006:** Outcomes in patients with CM.

Variables	Overall (*n* = 137)	Neri A (*n* = 68)	Neri B (*n* = 43)	Neri C (*n* = 15)	COIT (*n* = 11)	*p* Value
In hospital mortality, *n* (%)	25 (18.2)	5 (7.4)	11 (25.6)	6 (40.0)	3 (27.3)	0.006
ECMO, *n* (%)	15 (10.9)	3 (4.4)	7 (16.7)	4 (26.7)	1 (9.1)	0.045
New Stroke, *n* (%)	10 (7.3)	4 (5.9)	1 (2.3)	2 (13.3)	3 (27.3)	0.028
Revisit for bleeding, *n* (%)	17 (12.4)	5 (7.4)	6 (14.0)	3 (20.0)	3 (27.3)	0.193
Mediastinitis, *n* (%)	2 (1.5)	1 (1.5)	1 (2.3)	0 (0.0)	0 (0.0)	0.894
Paraplegia, *n* (%)	8 (5.8)	3 (4.4)	3 (7.0)	0 (0.0)	2 (18.2)	0.228
CRRT, *n* (%)	41 (29.9)	19 (27.9)	12 (27.9)	7 (46.7)	3 (27.3)	0.522
Tracheostomy, *n* (%)	6 (4.4)	4 (5.9)	1 (2.3)	0 (0.0)	1 (9.1)	0.558
Composite adverse events, *n* (%)	60 (43.8)	25 (36.8)	19 (44.2)	10 (66.7)	6 (54.5)	0.167
Ventilation time, d (median [IQR])	5.0 [3.0, 7.0]	5.0 [2.25, 7.0]	5 [3.0, 8]	6.0 [2.0, 12.0]	4.0 [2.0, 6.0]	0.902
ICU stay, d (median [IQR])	9.0 [6.0, 15.0]	9.0 [6.0, 16.75]	9 [5.0, 12.0]	10.0 [4.0, 17.0]	9.0 [6.0, 22.0]	0.920
Hospital stay, d (median [IQR])	21.0 [14.0, 30.0]	18.5 [14.0, 30.0]	22.0 [17.0, 32.0]	17.0 [10.0, 26.0]	25.0 [10.0, 34.0]	0.571

ECMO, extracorporeal membrane oxygenation; CRRT, continuous renal replacement therapy; ICU, intensive care unit. Composite adverse events = Revisit for bleeding + ECMO + Mediastinitis + Stroke + Paraplegia + CRRT + Tracheostomy + Mortality.

## Data Availability

Data are available upon request.

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
