# Peer review of "Coronary Malperfusion Secondary to Acute Type A Aortic Dissection: Surgical Management Based on a Modified Neri Classification"

_jcm, 2022, doi:10.3390/jcm11061693_

Round 1
Reviewer 1 Report
Review of article: ‘Coronary malperfusion secondary to acute type A aortic dissection: surgical management based on a modified Neri classification’ - J Clin Med
In their article, Tong et al. compare the results of repair of acute type A aortic dissection (ATAAD) in patients with and without coronary malperfusion (CM). They found that
surgical management based on a modified lesion classification achieved acceptable outcomes. The authors present an extensive experience in the treatment of ATAAD, over 1000 patients in a 5-year interval, and therefore this paper merits attention. However, there are some points which need to be clarified.
- They state that the number of patients with CM in their series is superior to that reported in the Literature. This may probably depend on patient selection which represents a major concern.
- Almost 50% of patients had type A CM which is the most common one and almost invariably found in ATAAD since in most cases the dissection extends in the non-coronary sinus and to the annulus but quite frequently it reaches also the origin of the right ostium. However, in such cases (type A) the dissection does not involve the lumen of the artery as shown in the drawing. Indeed, in the text there is no evidence on how many of these patients had really clinical signs of myocardial ischemia or if the diagnosis was only made on intraoperative findings.
- In hospital mortality of the entire CM cohort of patients is lower than 19%, very low considering that almost 30% of patients presented also with cardiogenic shock. Furthermore, the in hospital mortality of patients with Type A CM is lower than of “No CM” group (7.4% vs 7.8%) and the successful repair in this subset was greater than 98%.
These data suggest that Type A Neri classification can not be used alone to define and select patients with coronary malperfusion, unless clinical signs of malperfusion (myocardial ischemia) are recognized.
In my opinion, a better selection of patients should be done in order to avoid the risk to include patients without a real coronary malperfusion.
- It is also not clear which type of operation was made. It is indicated that most patients with type A had a successful repair and yet only 11% had a supracoronary ascending aorta replacement, 45% a root replacement (Bentall?). This is not clear: if you repair the coronary lesion why replace the root?
- 2% of patients had a Wheat procedure which in the text is described incorrectly as ‘aortic valve replacement and supracoronary ascending replacement’. However, I doubt you would perform a real Wheat procedure in ATAAD (see the designs in the original Wheat paper).
- It’s surprising that 100% of CM patients underwent arch replacement with FET procedure, that required a longer ACC time. I would expect that in patients with coronary malperfusion the goal was to reduce the ACC time as much as possible. Please comment.
Reviewer 2 Report
I have read with great interest the manuscripit “Coronary malperfusion secondary to acute type A aortic dissection: surgical management based on a modified Neri classification” by Tong G and colleagues. It is very impressive to include as many as 1018 patients with acute type A aortic dissection (ATAAD) among which 137 with coronary malperfusion (CM). This is a retrospective observational study that aimed at evaluating the incidence, management and outcomes of the patients with CM secondary to ATAAD. The authors also proposed some changes of Neri classification such as the introduction of new terminology, such as Coronary orifice intimal tear (COIT) as unique set of lesions. However, we found that some aspects merit further explanations and discussion.
First of all, myocardial ischemia after CM is considered a risk factor for postoperative in-hospital mortality as stipulated by Kawahito et al (ref 7) and Imoto et al (ref 9). The authors did not point out the actual prevalence, with or without ischemia, of their patients with CM. Is it possible to identify the patients with acute coronary ischemia due to CM? This is a concerning issue because CM with myocardial ischemia is strongly related to postrepair in-hospital mortality. Therefore, “acute coronary involvement” (ACI) may be a more appropriate term than “coronary malperfusion’’ for this special cohort because ACI secondary to ATAAD does not always cause myocardial ischemia.
Secondly, when discussing the outcome the classification into primary endpoints and secondary endpoints of the therapy may be useful for better understanding the text.
Looking forward to hearing from you,
Yours sincerely,
Reviewer 3 Report
I want to congratulate Tong et al. on the work and this manuscript.
The subject of the manuscript is relevant and encompasses a large patient population with acute type A aortic dissection. A recommendation for consideration by the authors. There are noted differences in table 6 as seen by the p-values. It might be of additional value, to show table 6 as an additional figure, displaying the differences in the different groups and the overall group.
Author Response
We thank the reviewer for the comment. We showed positive outcomes in table 6 with additional figure in our revision as suggested.